# Anti-selective [3+2] (Hetero)annulation of non-conjugated alkenes via directed nucleopalladation

Hui-Qi Ni[1], Ilia Kevlishvili[2], Pranali G. Bedekar[1], Joyann S. Barber[3], Shouliang Yang[3], Michelle Tran-Dubé[3], Andrew M. Romine [1], Hou-Xiang Lu[1], Indrawan J. McAlpine [3✉], Peng Liu [2✉] & Keary M. Engle [1✉]

2,3-Dihydrobenzofurans and indolines are common substructures in medicines and natural products. Herein, we describe a method that enables direct access to these core structures from non-conjugated alkenyl amides and *ortho*-iodoanilines/phenols. Under palladium(II) catalysis this [3 + 2] heteroannulation proceeds in an *anti*-selective fashion and tolerates a wide variety of functional groups. *N*-Acetyl, -tosyl, and -alkyl substituted *ortho*-iodoanilines, as well as free –NH$_2$ variants, are all effective. Preliminary results with carbon-based coupling partners also demonstrate the viability of forming indane core structures using this approach. Experimental and computational studies on reactions with phenols support a mechanism involving turnover-limiting, endergonic directed oxypalladation, followed by intramolecular oxidative addition and reductive elimination.

[1] Department of Chemistry, The Scripps Research Institute, 10550 North Torrey Pines Road, La Jolla, CA 92037, USA. [2] Department of Chemistry, University of Pittsburgh, 219 Parkman Avenue, Pittsburgh, PA 15260, USA. [3] Pfizer Oncology Medicinal Chemistry, 10770 Science Center Drive, San Diego, CA 92121, USA. ✉email: indrawan.mcalpine@pfizer.com; pengliu@pitt.edu; keary@scripps.edu

Catalytic (hetero)annulation[1–4] of C–C π-systems is a powerful method for constructing carbocyclic and heterocyclic core structures from comparatively simple progenitor compounds. Of special importance within the modern synthetic repertoire is the Pd(0)-catalyzed Larock indole synthesis, first described in 1991[5], which unites alkynes and 2-haloanilines (Fig. 1a). Later in 1995 and 2005, the Larock group reported analogous methods for benzo[b]furan and indene synthesis[6,7]. Mechanistically, such reactions are generally believed to proceed via oxidative addition, migratory insertion of the resultant arylpalladium(II) species, and C($sp^2$)–N/O/C reductive elimination.

While palladium-catalyzed alkyne annulation reactions are well developed and widely deployed in preparative chemistry, employment of alkenes in analogous processes is comparatively rare and limited to activated substrates like styrenes[8–10], 1,3-dienes[11], norbornenes[12], and enol ethers/esters[13–17] (Fig. 1b). Extension of this mode of reactivity to unactivated alkenes has proven to be challenging for several interrelated reasons: (1) competitive β-hydride elimination following migratory insertion; (2) inherently challenging C($sp^3$)–N/O/C reductive elimination[18]; and (3) competitive alkene isomerization pathways[19]. In 2015, the Jamison group reported an elegant method to access indolines from unactivated alkenes by taking advantage of nickel/photoredox dual catalysis[20], though the reaction required use of an N-acetyl protecting group and was only demonstrated with terminal alkenes, thereby limiting the product structures that can be accessed. To our knowledge, analogous processes between O-based and C-based coupling partners and unactivated alkenes remain unknown.

In order to bridge this important synthetic gap, we sought to develop an all-encompassing annulation process that could unite non-conjugated alkenes and a diverse collection of O-based, N-based, and C-based coupling partners. Specifically, we envisioned

that the issues highlighted above could potentially be circumvented by engaging an alternative mechanism, namely a sequence of Pd(II)-mediated Wacker-type nucleopalladation, followed by oxidative addition of the Ar–X bond to form a high-valent Pd(IV) intermediate, and finally C($sp^3$)–Ar reductive elimination. If successfully developed, such a process would potentially enable access to net anti-annulation, thereby complementing existing methods, which proceed in a syn-selective fashion. Of relevance to this proposal, the Mazet and Zhang groups have described Pd(0)/Pd(II) heteroannulation systems in which syn-oxy/aminopalladation is proposed to take place from a $L_n \cdot Pd^{II}(Ar)(X)$ intermediate, such that C($sp^3$)–O/N bond formation precedes C($sp^3$)–Ar bond formation[14–17].

During the past 5 years, our laboratories have developed a variety of transition-metal-catalyzed, three-component directed alkene 1,2-difunctionalization reactions facilitated by the 8-aminoquinoline (AQ) auxiliary, a strongly coordinating bidentate directing group. Using this strategy we and others have reported examples of hydrofunctionalization[21–33], dicarbofunctionalization[34–39], carboamination[40,41], and carbo-/aminoboration[42–45], among other transformations[46,47] (Fig. 1c). We recognized several issues that would need to be addressed to bring a directed (hetero)annulation process to fruition: (1) phenols and anilines have previously proven to be ineffective coupling partners in AQ-directed alkene additions in our hands, leading to the hypothesis that nucleopalladation in these cases is endergonic; (2) due to their intramolecular nature, the oxidative addition and reductive elimination transitions states involving Pd(IV) would likely be highly strained; and (3) competitive pathways arising from Heck-type or Wacker-type oxidative alkene addition or hydrofunctionalization could predominate. At the outset, our hypothesis was that the presence of an ortho-halogen would allow for rapid trapping of a potentially short-lived nucleopalladated intermediate.

In this work, by employing ambiphilic coupling partners we successfully achieve the (hetero)annulation of AQ-directed non-conjugated alkenes[48], thereby providing direct access to useful 2,3-dihydrobenzofurans, indolines, and indanes[49–52]. Mechanistic experiments and DFT studies shed light on the origins of diastereoinduction, anti-selectivity, and other aspects of this Pd(II)/Pd(IV) annulation process.

## Results

**Scope of phenols.** We began by using AQ-containing alkenyl amide **1a** as the model alkene and 2-iodophenol (**2aa**) as the coupling partner. After brief optimization (see Supplementary Information for details), we identified effective conditions using Pd(OAc)$_2$ (5 mol%) as the catalyst, K$_2$CO$_3$ (1.0 equiv) as the base, and HFIP (1.0 M) as the solvent, at 80 °C, under an air atmosphere, allowing for isolation of desired product **3aa** in 96% yield.

Having identified a high-yielding and operationally simple procedure, we next examined different ortho-iodophenols (Table 1). Phenols containing weakly electron-donating groups, namely 4-Me (**3ab**) and 4-$^t$Bu (**3ac**), furnished the corresponding products in 85% and 99% yield, respectively. 4-F (**3ad**), 4-Cl (**3ae**), 4-Br (**3af**), and 5-Br (**3ai**) phenols were also well tolerated and gave moderate to high yields, demonstrating the reactions compatibility with various halogen atoms. The presence of strongly electron-withdrawing groups at the 4-position, namely 4-CO$_2$Me (**3ag**) and 4-(4-CN-C$_6$H$_4$) (**3ah**), led to lower yields.

**Scope of anilines.** Next, we turned to evaluating the aniline scope (Table 1). With free NH$_2$ anilines, lowering the concentration to 0.5 M led to higher yields, and substrates with electron-

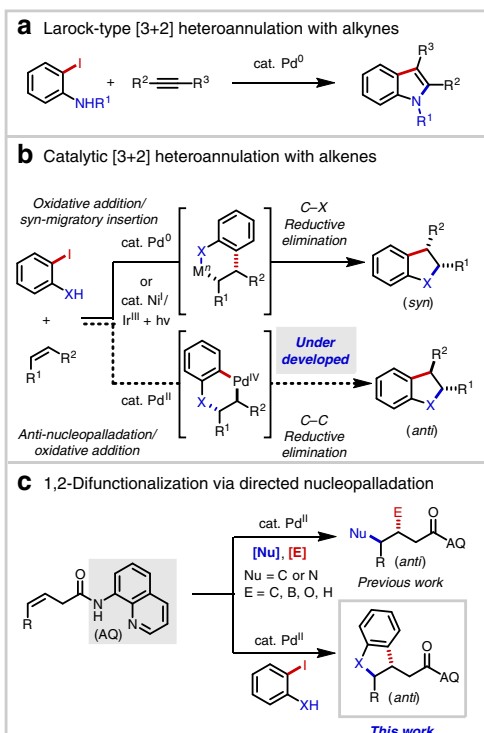

**Fig. 1 Background and project synopsis. a** Larock-type [3 + 2] heteroannulation with alkynes. **b** Catalytic [3 + 2] heteroannulation with alkenes. **c** 1,2-difunctionalization via directed nucleopalladation.

**Table 1 Scope of *ortho*-iodophenols, *ortho*-iodoanilines and carbon-based coupling partners.**

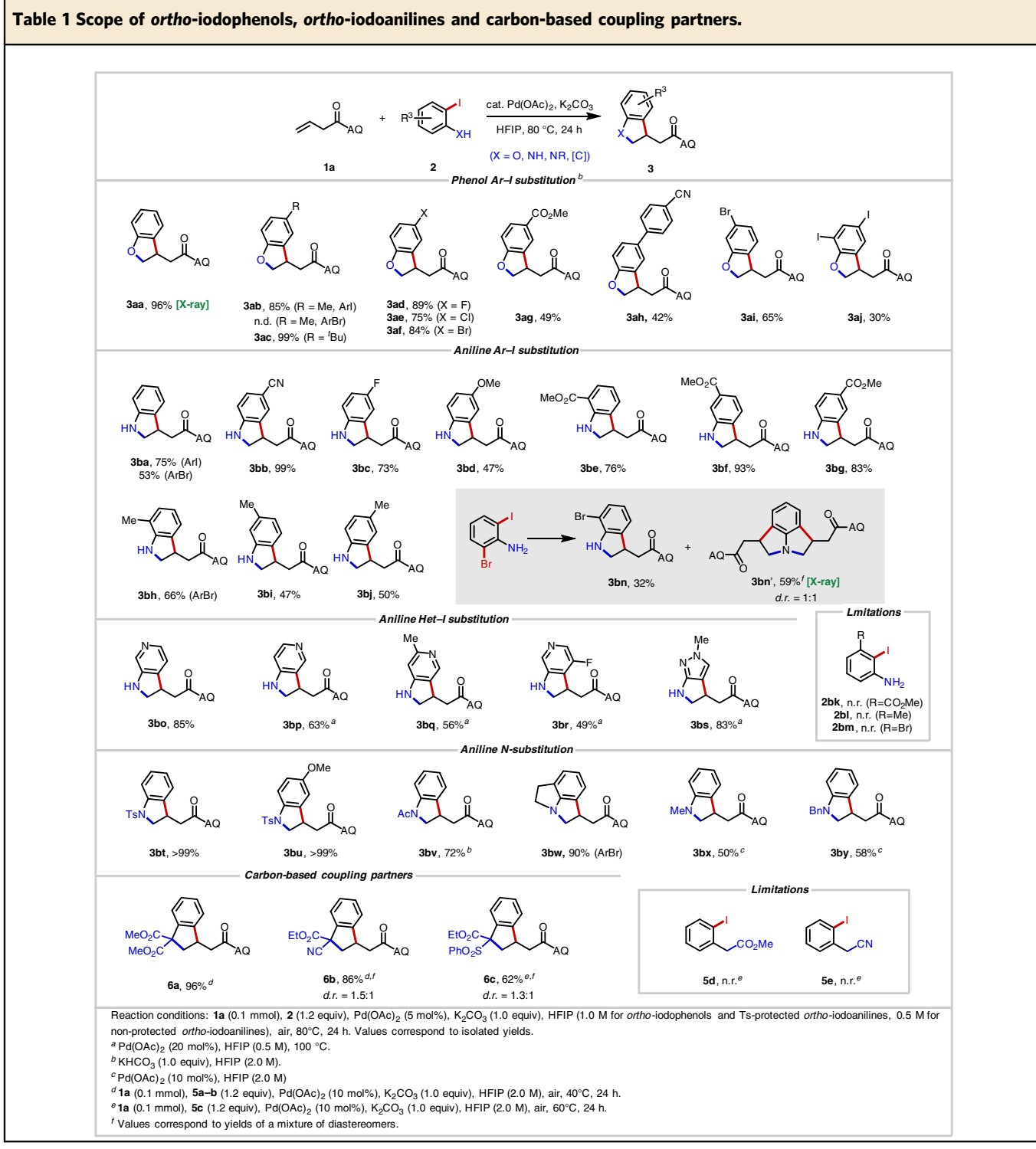

Reaction conditions: **1a** (0.1 mmol), **2** (1.2 equiv), Pd(OAc)$_2$ (5 mol%), K$_2$CO$_3$ (1.0 equiv), HFIP (1.0 M for *ortho*-iodophenols and Ts-protected *ortho*-iodoanilines, 0.5 M for non-protected *ortho*-iodoanilines), air, 80°C, 24 h. Values correspond to isolated yields.
[a] Pd(OAc)$_2$ (20 mol%), HFIP (0.5 M), 100 °C.
[b] KHCO$_3$ (1.0 equiv), HFIP (2.0 M).
[c] Pd(OAc)$_2$ (10 mol%), HFIP (2.0 M)
[d] **1a** (0.1 mmol), **5a–b** (1.2 equiv), Pd(OAc)$_2$ (10 mol%), K$_2$CO$_3$ (1.0 equiv), HFIP (2.0 M), air, 40°C, 24 h.
[e] **1a** (0.1 mmol), **5c** (1.2 equiv), Pd(OAc)$_2$ (10 mol%), K$_2$CO$_3$ (1.0 equiv), HFIP (2.0 M), air, 60°C, 24 h.
[f] Values correspond to yields of a mixture of diastereomers.

withdrawing groups performed better in this case. With *ortho*-iodoaniline, 75% of the desired product **3ba** was obtained, and pleasingly *ortho*-bromoaniline also provided product **3ba** in 53% yield. The ability to use bromoanilines is advantageous in many circumstances since they are easier to prepare and more readily available from commercial suppliers than the corresponding iodoanilines. 4-, 5-, and 6-CO$_2$Me-substituted 2-iodoanilines (**3be–3bg**) were all effective reaction partners, among which 5-CO$_2$Me (**3bf**) gave the highest yield of 93%. 4-CN-substituted 2-iodoaniline (**3bb**) delivered quantitative yield. Halogen atoms (**3bc**), alkyl groups (**3bi**, **3bj**), and electron-donating substituents

(**3bd**) on the aromatic ring led to decreased yields. With electron-donating groups, the product is prone to autooxidation to the corresponding indole, particularly in the case of **3bd**. 6-Me-substituted 2-bromoaniline (**3bh**) was also compatible in this reaction. 3-Substituted 2-iodoanilines were incompatible due to the steric hindrance (**2bk–2bm**).

Given the importance of azaindole derivatives as building blocks in medicinal chemistry[53,54], we next tested aminoiodopyridines. Notably, compared to azaindoles, the synthesis of azaindolines has been less thoroughly studied. We were wary that the basic and coordinating azine nitrogen atoms could pose a

**Table 2 Alkene scope.**

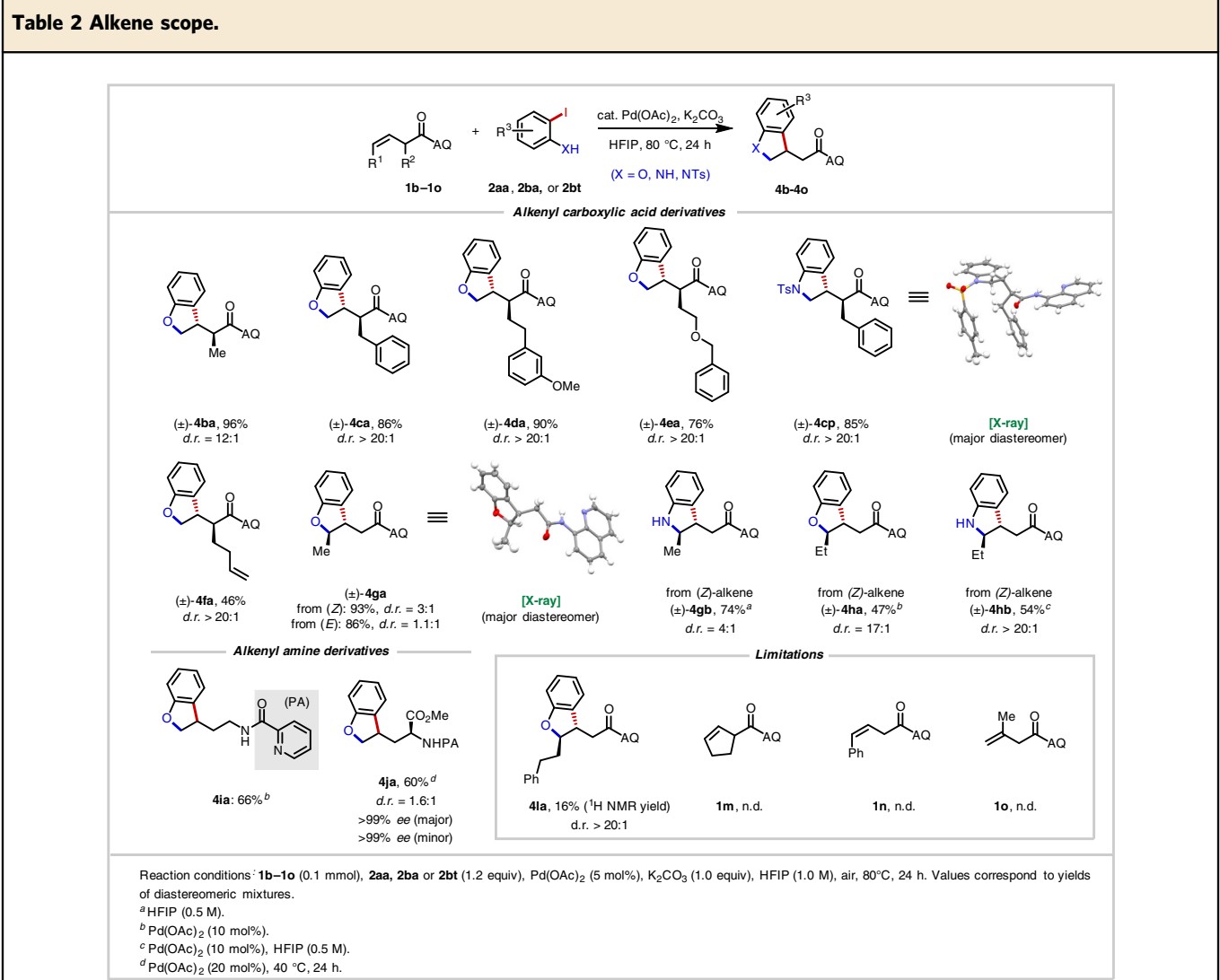

Reaction conditions: **1b–1o** (0.1 mmol), **2aa**, **2ba** or **2bt** (1.2 equiv), Pd(OAc)$_2$ (5 mol%), K$_2$CO$_3$ (1.0 equiv), HFIP (1.0 M), air, 80°C, 24 h. Values correspond to yields of diastereomeric mixtures.
[a] HFIP (0.5 M).
[b] Pd(OAc)$_2$ (10 mol%).
[c] Pd(OAc)$_2$ (10 mol%), HFIP (0.5 M).
[d] Pd(OAc)$_2$ (20 mol%), 40 °C, 24 h.

significant challenge, as is often the case in Pd(II)-catalyzed reactions[34,55]. To our delight, however, various pyridine-containing substrates were indeed compatible. For **3bo**, we obtained 85% yield under the standard conditions; for **3bp–3bs**, higher Pd loadings and higher temperatures were employed, but these also gave satisfactory yields. Of note we were able to prepare **3br**, which contains a fluorine atom at the sterically challenging site *ortho* to the reactive iodide group.

Anilines containing a variety of *N*-substituents were then tested, with different groups requiring different optimal reaction concentrations (see Supplementary Information). Tosyl (Ts) proved to be a highly effective protecting group. The yield of the simplest Ts-protected 2-iodoaniline was >99% (**3bt**). In cases where the product is susceptible to autooxidation (e.g., OMe-containing compound **3bd**), Ts protection leads to improved yield (>99% yield, **3bu**). Changing the protecting group to acetyl (Ac) (**3bv**) at higher concentration (2.0 M) gave a 72% yield. Tricyclic products (**3bw**) could also be formed in the reaction conditions, and we were intrigued to observe 59% of the 1:1 *cis/trans*-configured diaddition product **3bn′** along with 32% monoaddition product **3bn** when using 2-bromo-6-iodoaniline. Different *N*-alkyl 2-iodoanilines also proved to be effective in this reaction. By increasing the palladium loading and performing the reaction at higher concentration, *N*-Me (**3bx**) and *N*-Bn (**3by**) products were obtained in 50% and 58% yield, respectively.

**Scope of carbon-based coupling partners**. A small collection of carbon-based coupling partners were also compatible in this transformation, and the corresponding products **6a–6c** could be obtained in 62–96% yield with 10 mol% Pd(OAc)$_2$ and HFIP (2.0 M) (Table 1). Doubly activated pronucleophiles were required for efficient coupling (**5d** and **5e**), and in the case of coupling partners bearing two different electron-withdrawing groups (**6b** and **6c**), diastereoinduction was modest. This carboannulation process[56,57] enables preparation of indane structures that are found in numerous natural products and pharmaceutical compounds[58–60].

**Scope of alkenes**. We moved on to explore the alkene scope (Table 2). To our delight, reactions with α-substituted alkenyl amides proceeded efficiently and with high diastereoselectivity. Even with a small moiety, in this case a Me group (**4ba**), we obtained 96% yield and 12:1 d.r. With a larger group at the α-position (**4ca–4ea**), >20:1 d.r. and 76–90% yield were obtained. An X-ray crystal structure of the Ts-protected products **4cp** confirmed that the major product arises from C(sp$^3$)–Ar bond formation on the face of the alkene opposite of the α-substituent (*vide infra*). A diene substrate (**4fa**) reacted exclusively at the β,γ-alkene rather than the δ,ε-alkene, showcasing a unique example of chemoselectivity through substrate directivity. Notably, acyclic,

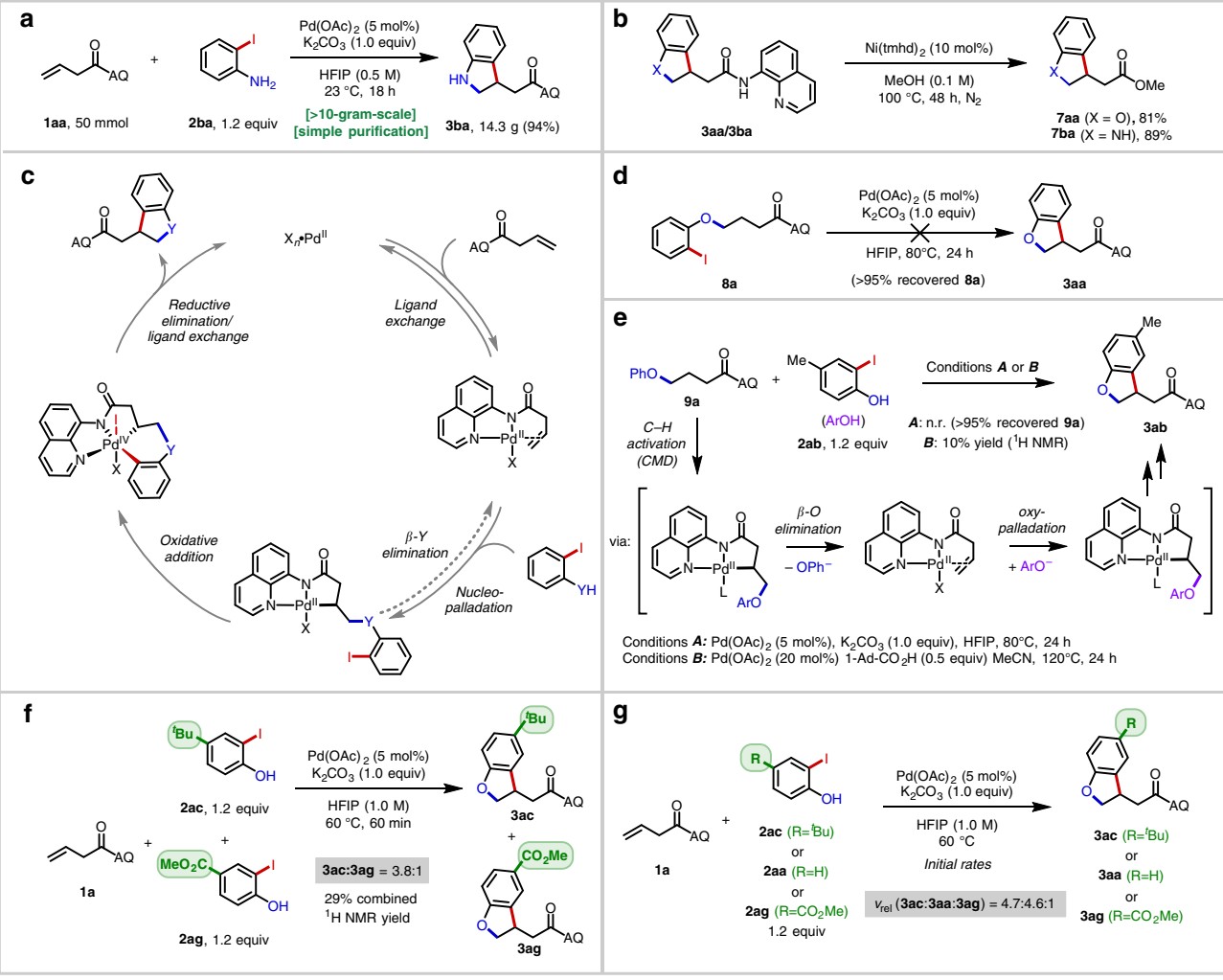

**Fig. 2 Large-scale reaction, directing group removal and mechanistic studies. a** 50 mmol-scale reaction. **b** Directing group removal. **c** Proposed mechanism. **d** Viability evaluation of an alternative pathway. **e** Evaluation of the potential reversibility of C(sp3)–O bond formation. **f** Competition experiments. **g** Initial rate studies.

non-conjugated internal alkenes, which are a difficult substrate class in nondirected systems due to issues with both reactivity and selectivity, engaged in productive [3 + 2] heteroannulation. Both (*Z*)-alkenes and (*E*)-alkenes were compatible, with (*Z*)-alkenes leading to higher d.r. (3:1 vs 1.1:1). According to previous studies of related transformations, internal alkenes are prone to *E/Z* isomerization (see Supplementary Information), and (*Z*)-alkenes react much faster in nucleopalladation than (*E*)-alkenes[29,43]. Both *ortho*-iodophenol (**4ga**, **4ha**) and *ortho*-iodoaniline (**4gb**, **4hb**) worked well for internal alkenes, and the X-ray crystal structure of **4ga** shows that the major product is in the *trans* configuration. Moreover, alkenyl amine substrates containing a 2-picolinyl amide (NHPA) directing group were also competent substrates. For the simple alkene and simple *ortho*-iodophenol (**4ia**), we isolated 66% yield by using 20 mol% Pd(OAc)2. Using an optically pure allylglycine starting material, we obtained 60% yield (**4ja**), 1:1.6 d.r. and >99% e.e without any enantiomerization by reducing the reaction temperature to 40 °C.

**Large-scale reaction and directing group removal.** This methodology could be conveniently scaled-up (Fig. 2a) and allowed for isolation of 14.3 g of **3ba** with minimal impurities after an operationally simple work-up and purification procedure (see Supplementary Information). Using the Morimoto–Ohshima

AQ amide alcoholysis method with catalytic Ni(tmhd)2[61], we obtained 81% (**7aa**) and 89% yield (**7ba**) yields for representative 2,3-dihydrobenzofuran and indoline products, respectively (Fig. 2b).

**Mechanistic studies.** In analogy to earlier work[34–47], we surmised that the reaction may proceed by the general mechanism depicted in Fig. 2c, involving a sequence of alkene coordination, nucleopalladation, intramolecular oxidative addition, and reductive elimination. To evaluate the viability of an alternative pathway comprised of alkene hydrofunctionalization followed by intramolecular C(sp3)–H arylation[62,63], we first performed a control experiment in which compound **8a** was subjected to the standard reaction conditions (Fig. 2d). In this case, no product was formed (with quantitative starting material recovery), establishing that **8a** is not a competent intermediate. This result is inconsistent with the two-step hydrofunctionalization/C(sp3)–H arylation mechanism. Next, to evaluate the potential reversibility of the C(sp3)–O bond formation step, we used compound **9a** as a starting material (Fig. 2e). Under the standard conditions, no reaction was observed. However, in the presence of 1-Ad-CO2H, which presumably facilitates C(sp3)–H activation via a concerted metalation/deprotonation (CMD) mechanism[64], we were able to observe small amounts of the [3 + 2] product **3ab**. The result

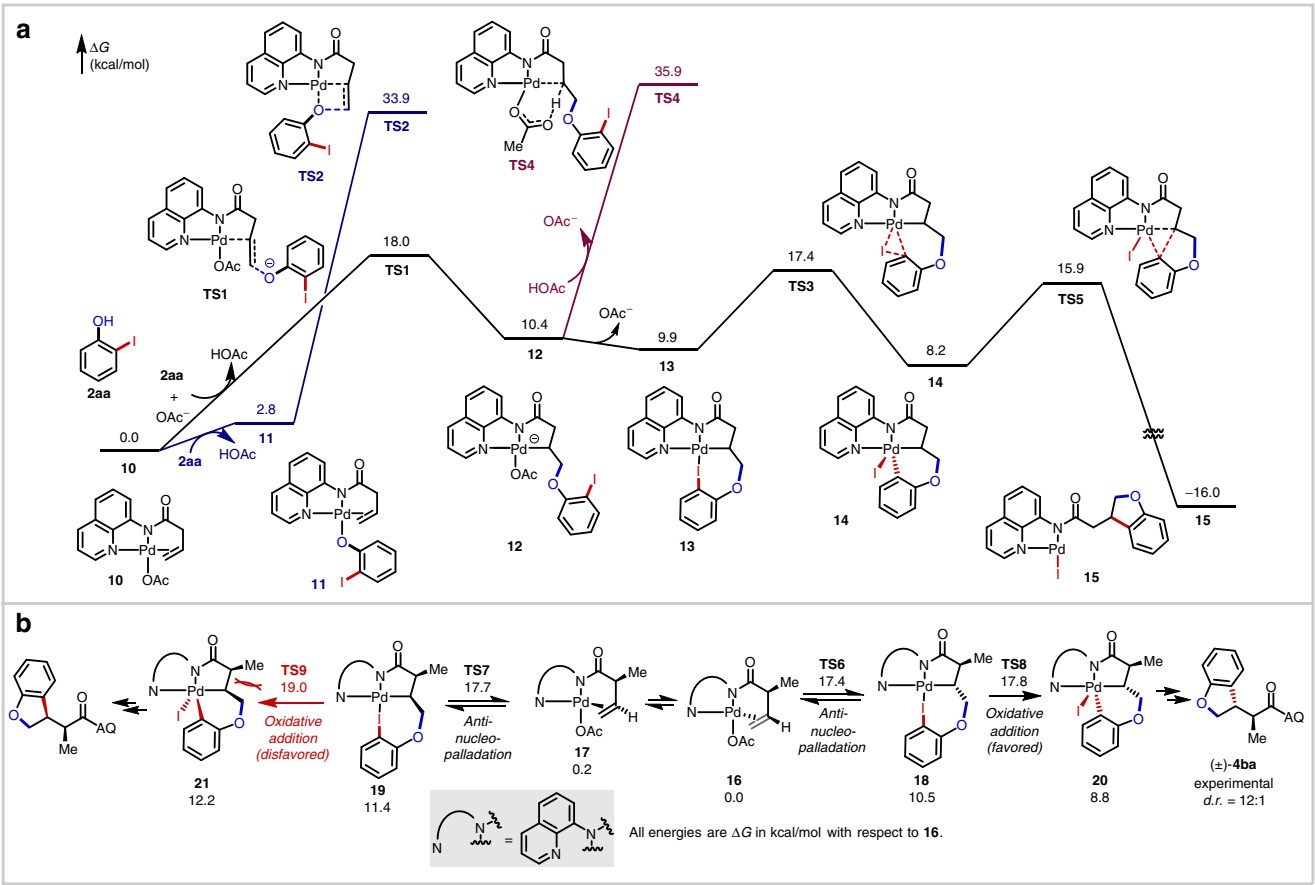

**Fig. 3 Computational studies. a** Computed energy profiles of the Pd-catalyzed [3 + 2] heteroannulation of **1a** and **2aa**. Calculations were performed using Gaussian 16 at the M06/SDD-6-311++G(d,p)/SMD(HFIP)//B3LYP-D3/SDD-6-31G(d) level of theory. **b** DFT calculations on the origin of diastereoselectivity with alkene **1b**.

suggests that C(sp³)–O bond formation is reversible from the alkylpalladium(II) species—at least when accessed through this alternative C(sp³)–H activation pathway. An analogous experiment with **8a** furnished an approximate 1:1 mixture of **3aa** and **3ab** in <10% combined yield (see Supplementary Information), consistent with the notion that C(sp³)–O bond formation is reversible and that oxidative addition and β-O elimination have similar energy barriers from the putative alkylpalladium(II) intermediate.

Next, a series of competition experiments were conducted. A one-pot experiment with electron-poor **2ag** and electron-rich **2ac** revealed the product-determining step is accelerated by electron-donating groups (Fig. 2f). This trend was recapitulated when global rates in single-component experiments were measured (Fig. 2g), which is consistent with either oxypalladation or C(sp³)–Ar reductive elimination (with the Ar fragment acting as the nucleophilic moiety[40]) as the turnover-limiting and product-determining steps for phenol substrates.

**Computational studies**. To clarify the reaction mechanism, density functional theory (DFT) calculations were performed (Supplementary Data 1) on the reaction of alkene **1a** and iodophenol **2aa** (Fig. 3a). The reaction starts with the complexation of the Pd(II) catalyst with **1a** to form π-alkene complex **10**. Several *syn*-nucleopalladation and *anti*-nucleopalladation pathways were evaluated computationally. The most favorable pathway proceeds through the *anti*-addition of deprotonated phenolate anion to **10** (via **TS1**), requiring a barrier of 18.0 kcal/mol. The competing

inner-sphere *syn*-addition pathway (**TS2**) from phenolate complex **11** is disfavored and requires a barrier of 33.9 kcal/mol. The palladacycle intermediate **12** is 10.4 kcal/mol less stable than **10**. The endergonicity is consistent with control experiments that indicate reversible nucleopalladation when this intermediate is accessed through an independent route (Fig. 2d). Dissociation of the acetate anion from **12** and the coordination of the *ortho*-iodo group to Pd forms complex **13**, which promotes the intramolecular oxidative addition (OA) (**TS3**, $\Delta G^{\ddagger} = 17.4$ kcal/mol). The resulting Pd(IV) complex **14** can undergo a relatively facile reductive elimination (**TS5**) with a barrier of 7.7 kcal/mol with respect to **14**. The competing protodepalladation of palladacycle **12** with acetic acid (**TS4**) is kinetically inaccessible, requiring an activation energy of 35.9 kcal/mol. In the overall annulation reaction from complex **10**, the nucleopalladation is the turnover-limiting step. Although the formation of palladacycle **12** is endergonic, the intramolecular oxidative addition/reductive elimination is entropically facilitated and proceeds without significant distortions in either transition state. It is notable that in this model reaction between **1a** and **2aa**, nucleopalladation and oxidative addition are computed to have very similar barriers (**TS1** vs. **TS3**). This raises the possibility that in reactions with other alkene substrates and coupling partners, the rate and selectivity could be controlled by either of these two steps (*vide infra*).

Last, we investigated the origin of diastereoselectivity in reactions with α-substituted alkenyl amides. We modeled the reaction of alkene **1b** bearing a small α-methyl substituent and iodophenol **2aa** (Fig. 3b). The two diastereomeric products are

formed via nucleopalladation of two nearly isoenergetic complexes **16** and **17** with two opposite π-faces of the alkene bound to the Pd. The nucleopalladation of **16** (**TS6**, $\Delta G^{\ddagger} = 17.4$ kcal/mol) is only slightly more favorable than that of **17** (**TS7**, $\Delta G^{\ddagger} = 17.7$ kcal/mol), due to the relatively early transition states. The diastereoselectivity is determined in the subsequent oxidative addition step (**TS8** vs. **TS9**), which requires higher activation energies than nucleopalladation for this substrate. The oxidative addition from palladacycle **19** (**TS9**) requires a 1.2 kcal/mol higher activation barrier than that from **18** due to the steric repulsion between the α-methyl substituent and the methylene, which are placed *cis* to each other. The predicted major diastereomeric product and the selectivity ($\Delta\Delta G^{\ddagger} = 1.2$ kcal/mol) are in agreement with the experimental results.

In conclusion, we have developed a highly versatile and selective method for [3 + 2] (hetero)annulation of non-conjugated alkenes via a directed nucleopalladation strategy. The transformation tolerates a diverse collection of *ortho*-iodophenols and *ortho*-iodoanilines, as well as α-(*ortho*-iodo-aryl)-carbonyl compounds, enabling access to substituted 2,3-dihydrobenzofurans, indolines, and indanes. This reaction is effective with challenging α-substituted alkenyl carbonyl substrates as well as alkenes bearing 1,2-dialkyl-substitution and proceeds in regioselective and diastereoselective fashion. The reaction tolerates both air and moisture and does not require any special precautions to perform. Mechanistic studies shed light on the origins of *anti*-selectivity, diastereoinduction, and other aspects of the mechanism of this Pd(II)/Pd(IV) annulation process.

## Methods

**General procedure for [3 + 2] (hetero)annulation**. To a 4-mL scintillation vial equipped with a Teflon-coated magnetic stir bar were added the Pd(OAc)$_2$ (0.005 mmol, 5 mol%), alkene substrate (0.1 mmol, 1 equiv), the appropriate coupling partner (0.12 mmol, 1.2 equiv), K$_2$CO$_3$ (0.1 mmol, 1 equiv), and HFIP (0.1 or 0.2 mL). The vial was sealed with a screw-top septum cap and placed in a heating block that was pre-heated to 80 °C. The reaction was allowed to run for 24 h and then was cooled to room temperature. The reaction mixture was filtered through a short plug of Celite, which was washed with EtOAc. The filtrate was concentrated under reduced pressure to afford a brown residue, which, upon purification by preparative TLC or column chromatography, afforded the pure product.

## Data availability

The data supporting the findings of this study are available within the article and its Supplementary Information. CCDC 1999888, 1999889, 2031197, and 2033387 contain the supplementary crystallographic data for this paper. These data can be obtained free of charge from The Cambridge Crystallographic Data Centre via www.ccdc.cam.ac.uk/data_request/cif.

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

## Acknowledgements

Dr. Gary J. Balaich, Dr. Milan Gembicky, Maximilian G. Bernbeck, and Dr. Jake B. Bailey (USCD) are acknowledged for assistance with X-Ray crystallography. We thank Prof. Zachary K. Wickens and his group at UW Madison for helpful discussion. This work was financially supported by the National Institutes of Health (5R35GM125052-03; 1R35GM128779), Pfizer, Inc., the Alfred P. Sloan Fellowship Program, the Camille Dreyfus Teacher-Scholar Program, and the Cottrell Scholars Program. Calculations were performed at the Center for Research Computing at the University of Pittsburgh and the Extreme Science and Engineering Discovery Environment (XSEDE) supported by the NSF. Dr. Jason S. Chen, Brittany B. Sanchez, and Emily J. Sturgell (Scripps Research Automated Synthesis Facility) are acknowledged for assistance with analysis and purification.

## Author contributions

H.-Q.N. and K.M.E. conceived the project. H.-Q.N. optimized the reaction conditions. H.-Q.N., P.G.B., J.S.B., S.Y., M.T.D., A.M.R., H.-X.L. examined the scope of the transformation. H.-Q.N. performed mechanistic experiments. I. K. and P.L. carried out DFT studies. I.J.M., P.L. and K.M.E. directed the research. H.-Q.N., I.K., P.L., and K.M.E. wrote the manuscript with input from all authors.

## Competing interests

The authors declare no competing interests.
