## [Peer Review File · Nature Communications]

REVIEWER COMMENTS

Reviewer #1 (Remarks to the Author):

McAlpine, Liu and Engle and coworkers describe the efficient synthesis of 2,3-dibenzofurans and indolines by a directed Pd-catalyzed heteroannulation of non-activated alkenes. The contribution is properly put into context, the important mechanistic concepts and potential flaws scholarly introduced. The study commences with a brief optimization of the reaction conditions, followed by delineation of the scope in O/N/C ambiphilic coupling partners. Alkene variations is described next and highlights the potential and originality of the present work. Specifically, the anti-selective nature of the 3+2 heteroannulation is revealed by variations of the substitution pattern of the unactivated alkene moiety. High dr have been obtained in certain cases. Overall, the scope is wide and limitations have been appropriately delineated. Control mechanistic experiments and DFT calculations have been carried out to shed light on a reasonable mechanism for the transformation. The origin of diastereoselectivity for one alkene has been investigated. The conclusions drawn are well supported by the data provided.

In conclusion, this is an excellent study, which requires only minor modifications (listed below) to reach the expected standards in the field and for publication in a journal of the level of Nature Communications.

Minor modifications/revisions/corrections:

*) p.2: "... for several interrelated reasons: 1) competitive β -hydride elimination following migratory insertion; (2) inherently challenging C(sp³)-N/O/C reductive elimination; [15] and (3) competitive alkene isomerization pathways." The authors are encouraged to provide references for arguments two and three.

*) p.3: change syn-oxy/aminopalladation is proposed to take place from a Ln•PdII(Ar)(X) intermediate, such that C(sp³)-O bond" to "syn-oxy/aminopalladation is proposed to take place from a Ln•PdII(Ar)(X) intermediate, such that C(sp³)-O/N bond".

*) p.5: change 'reaction condition' to 'reaction conditions'

*) The results obtained using 2-bromo-6-iodoaniline (3bn+3bn') indicate that 2-bromophenols are also compatible ambiphilic coupling patterns. First, the authors should explain why they did consider 2-iodophenols, which are significantly more expensive. Second, these results also suggest that oxidative addition is potentially reversible. The authors are encouraged to comment on this notion and could potentially explore it further by testing 2,6-diiodophenol and/or 2,4-diiodophenol (or the corresponding anilines) under the optimized conditions. For a relevant review, see: Nature Catalysis 2019, 2, 843.

*) p.7: 'with the chiral center in the product maintained'. The formulation is rather inaccurate and probably originates from a confusion between the notions of chirality and enantiomeric excess. The authors probably mean that there is no enantiomerization (i.e the reaction is enantiospecific). A chiral center would not be maintained if at least two of its substituents were to become identical.

*) p.7: 'clean impurity profile' sounds rather strange.

*) p.7 and Fig. 2-E: the proper abbreviation for adamantly is Ad not Ada.

*) Fig. 2: the legend does not match with the figure. 'A' has been used twice. 3 must be superscripted in sp³.

*) Fig. 2C: using 'X' on both the substrate and as a ligand on Pd might be a source of confusion. Use of 'X' and 'Y' is recommended.

*) Some limitations must appear in Table 2 (pick up some from S-39)

*) In the DFT study: did the authors consider an acetate assisted conversion of 11 into 12?

*) p.8 (2nd paragraph): ref 14a is ref 14

*) missing ref: Wolfe, J.P. Asian J. Org. Chem. 2017, 6, 636

Reviewer #2 (Remarks to the Author):

The manuscript by McAlpine, Liu, Engel and co-workers reports on the Pd(II)-catalyzed anti-selective [3+2] annulation of non-conjugated alkenes via directed oxa-, aza-, or carbo-palladation, by taking advantage of an 8-aminoquinoline directing group. Since the Engle group reported on the Pd(II)-catalyzed hydroamination of β,γ -unsaturated aliphatic amides with amines by taking advantage of an 8-aminoquinoline directing group in 2016 (ref 18), this 8-aminoquinoline-directed strategy has now been extended to various transformations including dicarbofunctionalization, carboamination, carbo-/aminoboration, and so on by the author's group, as well as other groups. In a current report, the authors extended the reaction to the (hetero)annulation of β,γ -unsaturated aliphatic amides with 2-iodoanilines or 2-iodophenol to provide indoline or benzofuran derivatives. The present reaction proceeds in an anti-selective manner and tolerates a wide variety of important functional groups. The authors also conducted preliminary mechanistic experiments and DFT calculations in an attempt to understand the mechanism for the reaction. The results show that reaction involves a sequence involving alkene coordination, nucleopalladation, intramolecular oxidative addition of a C-I bond, and reductive elimination. The results of a mechanistic study also shed light on the origin of the anti- and diastereo-selectivity. This reviewer recommends that the paper be accepted for publication in Nature Communications, after the revisions listed below are addressed.

- (1) In page 2, second paragraph, the authors stated that an analogous process between O- and C-based coupling partners and non-conjugated alkenes remains unknown. However, this is not true. Catellani reported on a Pd-catalyzed reaction using norbornene as an alkene (ref 11). In the above sentence, "non-conjugated alkenes" should be changed to "unactivated alkenes" to reflect this fact.
- (2) Zhang, Liu, and coworkers recently reported on the Pd-catalyzed reaction of 2-bromoaniline and 2-bromophenol derivatives with 2,3-dihydrofurans (Chem. Sci. 2020, 11, 6283). This paper should be cited.
- (3) Page 4, second paragraph: 3be-3bf should be 3be-3bg.
- (4) Concerning the control experiment shown in Fig 2D, Chen and coworkers reported on the Pd(II)-catalyzed annulation of benzo-rings by the intramolecular coupling of an aryl iodide and a C(sp³)-H bond, by taking advantage of an 8-aminoquinoline directing group (Org. Lett. 2010, 12, 3414). This paper should be cited.
- (5) The authors examined various reaction parameters for optimizing the reaction conditions in detail (Table S3-S7). However, only the yields of the expected products were shown. It would be informative for readers if the yields of by-products, such as hydrofunctionalization products are also given. Or did the authors recover the unreacted substrate?
- (6) ¹H NMR data: Five protons are missing in the assignment of 2bu. One proton is missing in 3bj, 3bc, 3bf, 3be, 3bi, 3bh, 3bs and (\pm)-4bh. Four protons are missing in (\pm)-4gb.
- (7) 3bd: One proton is missing and [M+H]⁺ should be C₂₀H₂₀N₃O₂, not C₂₀H₁₉N₃O₂. The same thing is observed in the cases of 3bn, 3bn', 3bp, 3bo, 3bw and 7ba. Please check them carefully.

REFEREE 1

Recommendation: Revisions required

1. Commenting on a portion of the introduction that reads “for several interrelated reasons: 1) competitive β -hydride elimination following migratory insertion; (2) inherently challenging C(sp³)-N/O/C reductive elimination;^[15] and (3) competitive alkene isomerization pathways,” the reviewer writes: “The authors are encouraged to provide references for arguments two and three.”

Response: Regarding reason (2), reference 18 has been cited to substantiate the statement that C(sp³)-N/O reductive elimination from a Pd(II) center is a challenging/high-energy process. Regarding reason (3), we note that *E/Z*-isomerization was indeed observed in this reaction system under certain reaction conditions. This data has now been added to the Supporting Information (see Tables S3–5 for more details). To further address this point a review about remote functionalization through alkene isomerization has been cited (Ref. 19).

2. The reviewer suggests that we change “syn-oxy/aminopalladation is proposed to take place from a L •Pd (Ar)(X) intermediate, such that C(sp³)-O bond” to “syn-oxy/aminopalladation is proposed to take place from a L •Pd (Ar)(X) intermediate, such that C(sp³)-O/N bond”.

Response: We changed “C(sp³)-O” to “C(sp³)-O/N”

3. The reviewer suggested that we change “reaction condition” to “reaction conditions”

Response: We changed “this reaction condition” to “these reaction conditions”

4. The reviewer writes, “The results obtained using 2-bromo-6-iodoaniline (3bn+3bn’) indicate that 2-bromophenols are also compatible ambiphilic coupling patterns. First, the authors should explain why they did consider 2-iodophenols, which are significantly more expensive. Second, these results also suggest that oxidative addition is potentially reversible. The authors are encouraged to comment on this notion and could potentially explore it further by testing 2,6-diiodophenol and/or 2,4-diiodophenol (or the corresponding anilines) under the optimized conditions. For a relevant review, see: Nature Catalysis 2019, 2, 843.”

Response: In an effort to address this comment, we attempted a model reaction with 4-methyl-2-bromophenol, and in this case no desired product was observed under the standard conditions. (This result has been added to the revised version of Table 1).

In regards to the second point about the reversibility of C(sp²)-I oxidative addition, we note that there is a fundamental difference between the examples in the cited review and the present work, because the examples covered in the review generally involve Pd(0)/Pd(II) catalysis, whereas the present system likely involves a Pd(II)/Pd(IV) redox couple. In the Pd(0)/Pd(II) systems both the 2- and 6-C-X presumably have similar rates of oxidative addition to Pd(0) catalyst, and the C-X bonds in starting material and product are both expected to be reactive. Hence, the detection of C-X-bond-containing product rules out an *irreversible* C-X oxidative addition. The same logic does not necessarily apply here. The free Pd(II) catalyst is not expected to have any background reaction rate with the aryl iodide, since the Pd(II)/Pd(IV) redox couple requires a stronger oxidant. Only upon formation of the alkylpalladium(II) intermediate (upon nucleopalladation) is this oxidative addition expected to be viable. Hence, the detection of a C-X bond in the product does have the same mechanistic ramifications. Nevertheless, we were curious about this suggestion, so we subjected 2,4,6-triiodophenol, and we were able to detect 30% product with 5% catalyst. This result has been added to Table 1, and the Supporting Information has been updated.

5. Commenting on a portion of the text that reads, “with the chiral center in the product maintained,” the reviewer writes: “The formulation is rather inaccurate and probably originates from a confusion between the notions of chirality and enantiomeric excess. The authors probably mean that there is no enantiomerization (i.e the reaction is enantiospecific). A chiral center would not be maintained if at least two of its substituents were to become identical.”

Response: Yes, what we mean is that there is no enantiomerization. We changed “with the chiral center in the product maintained” to “without any enantiomerization”.

6. The reviewer notes, “‘clean impurity profile’ sounds rather strange.”

Response: We changed “clean impurity profile” to “with minimal impurities”

7. The reviewer notes that on p.7 and Fig. 2-E, the proper abbreviation for adamantyl is Ad not Ada.

Response: We changed “Ada” to “Ad” in both main manuscript and SI.

8. The reviewer writes, “Fig. 2: the legend does not match with the figure. ‘A’ has been used twice. 3 must be superscripted in sp³.”

Response: These problems were fixed.

9. The reviewer writes, “Fig. 2C: using ‘X’ on both the substrate and as a ligand on Pd might be a source of confusion. Use of ‘X’ and ‘Y’ is recommended.”

Response: We changed “X” on substrate to “Y”.

10. The reviewer writes, “Some limitations must appear in Table 2 (pick up some from S-39).”

Response: Selective limitations were added to Table 2.

11. The reviewer asks, “In the DFT study: did the authors consider an acetate assisted conversion of 11 into 12?”

Response: We updated the 3D and chemdraw structure for **TS10** (Figure S7), which involves *syn* addition of deprotonated phenol substrate to the olefin to better highlight inner sphere character and some interaction between the palladium and oxygen in the axial position in this transition state, which would correspond to acetate assisted conversion of 11 directly into 12. We have also calculated another transition state (**TS15**, Figure S7) in which, acetate binds Pd center in the axial position, for which the transition state is highly reminiscent of the inner sphere *syn* addition process (**TS2**). Acetate binding is disfavored and the TS requires 42.5 kcal/mol activation energy.

12. The reviewer notes that on p.8 (2nd paragraph): ref 14a should be updated.

Response: We changed ref 14a to ref 40.

13. The reviewer suggests that we add the following reference: Wolfe, J.P. Asian J. Org. Chem. 2017, 6, 636

Response: We added this reference. (Ref.4)

REFeree 2

Recommendation: Revisions required

1. The referee writes: “In page 2, second paragraph, the authors stated that an analogous process between O- and C-based coupling partners and non-conjugated alkenes remains unknown. However, this is not true. Catellani reported on a Pd-catalyzed reaction using norbornene as an alkene (ref 11). In the above sentence, ‘non-conjugated alkenes’ should be changed to ‘unactivated alkenes’ to reflect this fact.

Response: We changed “non-conjugated alkenes” to “unactivated alkenes”.

2. The reviewer writes, “Zhang, Liu, and coworkers recently reported on the Pd-catalyzed reaction of 2-bromoaniline and 2-bromophenol derivatives with 2,3-dihydrofurans (Chem. Sci. 2020, 11, 6283). This paper should be cited.”

Response: We have added Refs. 16 and 17.

3. The referee notes that on Page 4, second paragraph: 3be-3bf should be 3be-3bg.

Response: We changed “**3be–3bf**” to “**3be–3bg**”.

4. The referee writes: “Concerning the control experiment shown in Fig 2D, Chen and coworkers reported on the Pd(II)-catalyzed annulation of benzo-rings by the intramolecular coupling of an aryl iodide and a C(sp³)-H bond, by taking advantage of an 8-aminoquinoline directing group (Org. Lett. 2010, 12, 3414). This paper should be cited.”

Response: This is already cited (Ref. 63).

5. The reviewer writes, “The authors examined various reaction parameters for optimizing the reaction conditions in detail (Table S3–S7). However, only the yields of the expected products were shown. It would be informative for readers if the yields of by-products, such as hydrofunctionalization products are also given. Or did the authors recover the unreacted substrate?”

Response: To address this concern, we updated Tables S3–S5. When different solvents, concentrations, and organic bases (*e.g.*, Et₃N) were tested, the major byproduct was the isomerized alkene. Some unreacted substrate was also observed when different bases were screened.

6. Regarding the ¹H NMR data, the reviewer writes: “Five protons are missing in the assignment of 2bu. One proton is missing in 3bj, 3bc, 3bf, 3be, 3bi, 3bh, 3bs and (±)-4bh. Four protons are missing in (±)-4gb.”

Response: These problems have been fixed. Generally the missing protons in these cases are from the broad NH peaks. Now they have been added manually in the text, but sometimes they cannot be detected after the baseline correction.

7. Regarding the SI, the reviewer notes: “One proton (for compound **3bd**) is missing and [M+H]⁺ should be C₂₀H₂₀N₃O₂, not C₂₀H₁₉N₃O₂. The same thing is observed in the cases of 3bn, 3bn’, 3bp, 3bo, 3bw and 7ba. Please check them carefully.”

Response: These problems have been fixed.

REVIEWERS' COMMENTS

Reviewer #1 (Remarks to the Author):

The authors have addressed all my concerns very seriously and rigorously (including by running additional experiments and performing complementary DFT calculations). They have done more than the bare minimum and they should be commended for their professionalism. The manuscript has now reached the standards of excellence set by Nature Communications and is therefore publishable in its current version.

Reviewer #2 (Remarks to the Author):

The authors have incorporated all of my suggestions into their revised manuscript and this referee now concludes that this paper is now acceptable for publication in Nature Communicatios.